# Resampling methods for class imbalance in clinical prediction models: A scoping review protocol

Osama Abdelhay[1]*, Adam Shatnawi[2], Hassan Najadat[2], Taghreed Altamimi[3]

1 Department of Data Science and Artificial Intelligence, Princess Sumaya University for Technology, Amman, Jordan, 2 Department of Computer Science, Jordan University of Science and Technology, Irbid, Jordan, 3 College of Engineering and Advanced Computing, Alfaisal University, Riyadh, Saudi Arabia

* o.abdelhay@psut.edu.jo

## Abstract

### Introduction

Class imbalance—where clinically important "positive" cases make up less than 30% of the dataset—systematically reduces the sensitivity and fairness of medical prediction models. Although data-level techniques, such as random oversampling, random undersampling, SMOTE, and algorithm-level approaches like cost-sensitive learning, are widely used, the empirical evidence on when these corrections improve model performance remains scattered across different diseases and modelling frameworks. This protocol outlines a scoping systematic review with meta-regression that will map and quantitatively summarise 15 years of research on resampling strategies in imbalanced clinical datasets, addressing a key methodological gap in reliable medical AI.

### Methods and analysis

We will search MEDLINE, EMBASE, Scopus, Web of Science Core Collection, and IEEE Xplore, along with grey literature sources (medRxiv, arXiv, bioRxiv) for primary studies (2009–31 Dec 2024) that apply at least one resampling or cost-sensitive strategy to binary clinical prediction tasks with a minority-class prevalence of less than 30%. There will be no language restrictions. Two reviewers will screen records, extract data using a piloted form, and document the process in a PRISMA flow diagram. A descriptive synthesis will catalogue the clinical domain, sample size, imbalance ratio, resampling strategy, model type, and performance metrics where 10 or more studies report compatible AUCs. A random-effects mixed-effects meta-regression (logit-transformed AUC) will be used to examine the effect of moderators, including imbalance ratio, resampling strategy, model family, and sample size. Small-study effects will be assessed with funnel plots, Egger's test, trim-and-fill, and weight-function models; influence diagnostics and leave-one-out analyses will

**Data availability statement:** No datasets were generated or analysed during the current study. All relevant data from this study will be made available upon study completion.

**Funding:** The author(s) received no specific funding for this work.

**Competing interests:** The authors have declared that no competing interests exist.

evaluate robustness. Since this is a methodological review, formal clinical risk-of-bias tools are optional; instead, design-level screening, influence diagnostics, and sensitivity analyses will enhance transparency.

## Discussion

By combining a comprehensive conceptual framework with quantitative estimates, this review aims to determine when data-level versus algorithm-level balancing leads to genuine improvements in discrimination, calibration, and cost-sensitive metrics across various medical fields. The findings will help researchers select concise, evidence-based methods for addressing imbalance, inform journal and regulatory reporting standards, and identify research gaps such as the under-reporting of calibration and misclassification costs, which must be addressed before balanced models can be reliably trusted in clinical practice.

### Systematic review registration

INPLASY202550026.

---

### Introduction

Medical prediction datasets often exhibit an imbalance, with the clinically important "positive" class making up less than 30% of observations. This skew systematically biases traditional (e.g., logistic regression) and modern machine-learning classifiers towards the majority class, reducing sensitivity for the minority group [1–3].

To mitigate this threat, a set of data-level resampling strategies—random oversampling (ROS), random undersampling (RUS), and the Synthetic Minority Oversampling Technique (SMOTE)—modifies the training data before modelling [1,4,5]. Although commonly used, ROS can cause overfitting due to duplicate instances, RUS may discard potentially informative data points, and SMOTE or its variants might generate unrealistic synthetic examples [6–9].

Evidence comparing resampling with alternative strategies remains inconclusive. An extensive systematic review showed no consistent performance advantage of machine learning over logistic regression when event-per-variable ratios were adequate [10]. Furthermore, simulation and empirical studies suggest that effective sample size planning, rather than aggressive post-hoc balancing, often negates the need for resampling [11–16].

At the algorithm level, cost-sensitive learning directly penalises errors in the minority class and can outperform methods that operate at the data level; however, it is infrequently reported in medical AI research [4,17].

Developments in binary classification theory—from early statistical formulations to perceptrons, support vector machines, and boosted ensembles—highlight how model choice interacts with class distribution and cost structure [18–21].

Across clinical class-imbalance settings, approaches span data-level resampling (random over/undersampling; SMOTE and variants), algorithm-level/cost-sensitive

learning, and increasingly ensembles/transfer learning. Resampling can be helpful when minority events are scarce, but it may induce boundary distortion/overfitting. Cost-sensitive methods align optimisation with misclassification costs. Ensembles and transfer learning can improve robustness but add complexity and computational demands. Given heterogeneity in prevalence, thresholds, and reporting, we restrict quantitative pooling to ROC-AUC and synthesise PR-AUC, MCC, F1, calibration, and decision-analytic measures descriptively, with PR-AUC/MCC receiving greater interpretive weight under skew. A fuller comparative appraisal (pros/cons and clinical suitability) is deferred to the results paper, consistent with protocol scope [22–24].

In this context, we will undertake a scoping systematic review with meta-regression to (i) map the resampling and cost-sensitive strategies employed in imbalanced medical datasets, (ii) quantify their effects on discrimination and calibration, and (iii) identify methodological moderators and research gaps. This protocol outlines the intended methods.

## Objectives

**Primary objective.** This study aims to assess whether, in clinical prediction studies with binary outcomes and a minority-class prevalence below 30%, applying data-level resampling or algorithm-level cost-sensitive strategies significantly improves model performance compared to training on the original imbalanced data.

**Specific objectives.**

1. **Evidence mapping** – Catalogue the complete range of imbalance correction strategies, including oversampling, undersampling, hybrids, and weighted or focal-loss models, reported between 2009 and 2024. Also include the clinical domains, dataset sizes, imbalance ratios, and modelling frameworks for these strategies.

2. **Comparative effectiveness** – Quantify and compare discrimination metrics (e.g., AUC, sensitivity, specificity) and, where available, the calibration metrics achieved by

   ◦ oversampling,

   ◦ undersampling,

   ◦ hybrid pipelines, and

   ◦ cost-sensitive algorithms,

   against models trained without any balancing.

3. **Moderator analysis**—Employing mixed-effects meta-regression, evaluate how study-level characteristics (imbalance ratio, sample size, number of predictors, model family, and clinical domain) impact the effectiveness of each imbalance-correction strategy.

4. **Assess bias and robustness** by examining the effects of small studies, publication bias, and significant outliers through funnel-plot diagnostics, trim-and-fill, weight-function models, and leave-one-out analyses; assess how these factors affect pooled estimates.

5. **Methodological gap identification** – Emphasise recurring pitfalls, such as neglecting calibration, misclassification costs, or external validation, and develop evidence-based recommendations for future research and reporting.

We hypothesise: (H1) Conditional on adequate sample size, resampling strategies (over/under/hybrid/SMOTE-type) do not improve predictive performance over no resampling in imbalanced binary clinical prediction tasks. (H2) Cost-sensitive methods outperform pure over/undersampling at IR < 10%; (H3) Hybrid (resampling+algorithmic) methods outperform single-strategy approaches; (H4) External validation yields lower AUC than internal; (H5) Studies reporting calibration perform better on net benefit where available." (Exploratory if data sparse.);

Covariates: imbalance ratio, sample size, validation tier, clinical domain, leakage safeguards.

## Methods

This protocol adheres to the PRISMA-P (S3 file) [25] and PRISMA-ScR [26] guidelines and has been registered with INPLASY (ID: INPLASY202550026) (S1 File). Any amendments will be recorded in the INPLASY record. Amendments (e.g., eligibility or analysis changes) will be logged with date, rationale, and impacted sections in a public registry (INPLASY/OSF) and cited in the final report.

### Eligibility criteria (PICOTS)

- **Population:** Clinical prediction studies that analyse binary outcomes with an explicit minority-class prevalence of less than 30%. For this review, a binary outcome is limited to diagnostic, prognostic, or treatment-response predictions in which the dependent variable has exactly two mutually exclusive states (e.g., disease present/absent).

- **Interventions:** Data-level resampling (random oversampling, random undersampling, SMOTE or variants, hybrid pipelines) and algorithm-level cost-sensitive strategies (weighted losses, focal loss).

- **Comparators:** Models trained on the original imbalanced data and/or alternative resampling or weighting strategies.

- **Outcomes:** Primary—AUC; secondary—sensitivity, F1-score, specificity, balanced accuracy, calibration metrics, and reported mis-classification costs.

- **Timing:** Publications from 1 Jan 2009–31 Dec 2024.

- **Study design** includes retrospective or prospective primary studies (such as model-development and validation papers) and systematic reviews that reanalyse primary data. Excluded are simulation-only papers, non-binary tasks, or abstracts that lack methods. Studies focusing solely on radiomics, image-segmentation pipelines, or pixel-level classification tasks will also be excluded, as these do not produce patient-level binary predictions.

- **Scope exclusion (imaging segmentation/radiomics):** We exclude pixel/voxel-level **segmentation** and radiomics tasks because they optimise **dense, pixel-level predictions** and are evaluated with **overlap/shape metrics** (e.g., Dice/Jaccard/Hausdorff), which are not commensurable with **patient-level clinical prediction** (e.g., ROC-AUC, PR-AUC, calibration) that is the focus of this review. Including segmentation would mix fundamentally different targets, class-imbalance structures, and metrics; therefore, such studies are out of scope. [27,28]

### Information sources and search strategy

Searches will be conducted in MEDLINE (PubMed), EMBASE, Scopus, Web of Science Core Collection, and IEEE Xplore. A peer-reviewed strategy combines controlled vocabulary and free-text terms to address class imbalance, resampling, and clinical prediction; an example MEDLINE string is provided in the S2 File. No language limits were applied, but non-English full texts had to be translatable.

**Grey literature (medRxiv/arXiv/bioRxiv/GitHub):** We include medRxiv, arXiv, bioRxiv, and GitHub to (i) reduce publication bias/small-study effects by capturing studies not yet in indexed journals, as recommended by major evidence-synthesis guidance, and (ii) map rapidly evolving ML methods whose earliest public disclosure is often via preprints/code releases. To mitigate risks (variable peer review/reporting quality), we apply minimum reporting standards (TRIPOD+AI-aligned task clarity, data splits/leakage safeguards, model specification, performance reporting, and reproducibility) and versioning (latest preprint version; tagged GitHub commit). We will (a) label preprints/code-only sources explicitly, (b) exclude records failing minimum standards from the synthesis (retaining them in the PRISMA flow), and

(c) run sensitivity analyses that exclude grey-literature records to assess their influence on conclusions. This approach follows PRISMA/PRISMA-ScR, which aims to map evidence while comprehensively managing transparently reporting quality.

We will screen these sources, but include a record in synthesis only if the minimum reporting is met:

1. **Predictive task clarity** (target population, outcome definition, prediction horizon).

2. **Data & split transparency** (source, inclusion/exclusion, train/validation/test strategy; leakage safeguards);

3. **Model specification** (algorithms, hyperparameters, resampling/cost strategies);

4. **Performance reporting** aligned with TRIPOD+AI (discrimination; threshold-dependent metrics when used; calibration if available) and, for imaging-AI studies, CLAIM elements as applicable;

5. **Reproducibility** (accessible code or sufficient procedural detail to replicate). Records failing these are catalogued but excluded from synthesis (retained in PRISMA flow). [29,30]

**Version control for preprints/GitHub:** For preprints, we use the latest version at extraction. For GitHub, we require a tagged release/commit hash to ensure reproducibility. Reporting items and reproducibility checks for grey-literature records are aligned with TRIPOD+AI, where LLM-based prediction studies appear; TRIPOD-LLM items will be consulted when applicable. [31]

## Study selection

Search results will be imported into Zotero for deduplication [32] and prioritised with ASReview [33]. Two reviewers will independently screen titles and abstracts, followed by full texts, resolving conflicts by consensus or through third-party adjudication. Reasons for exclusion will be recorded and displayed in a PRISMA flow diagram [25]. Data missing from the full text will be requested from authors (two-week window). We will detect duplicate/overlapping cohorts (e.g., pre-print→journal of the same dataset) by matching data sources/time windows/outcomes and will retain the most complete, peer-reviewed record; secondary records contribute unique methodological details. A de-duplication table will document decisions. [34]

## Data extraction

A standardised, pilot-tested form will record bibliometrics, clinical domain, sample size, imbalance ratio, resampling strategy, model family, performance metrics, calibration statistics, and cost-sensitive measures. Two independent reviewers will extract all items twice into a REDCap database (version 14.0.19). A third reviewer will run the comparison report, resolve discrepancies, and export a single verified dataset. Statistical analyses will be performed in R (v4.4.0) using the metafor (v4.8-0), dplyr (v1.1.4), and ggplot2 (v3.5.2) packages. After publication, all code and a session-info file will be uploaded to the OSF repository.

## Outcomes and effect measures

### a. Evaluation Metrics

Why is accuracy insufficient? In imbalanced settings, accuracy can be high while the minority class is poorly detected; we report it only for completeness.

Discrimination. We prioritise ROC-AUC for quantitative pooling due to ubiquity and cross-study comparability, while noting its optimistic behaviour under skewed prevalence. PR-AUC will be emphasised in interpretation because it reflects positive-class performance and is more informative in cases of imbalance. [35]

Threshold-dependent metrics: We will tabulate/visualise F1, sensitivity/specificity, and Matthew's correlation coefficient (MCC); MCC provides a balanced assessment from the full confusion matrix and often outperforms accuracy/F1 in skewed data. These metrics will not be pooled because thresholds and prevalences vary across studies [36].

Calibration & decision impact: Calibration slope/intercept and Brier score will be summarised descriptively (no pooling). Where authors report decision-curve analysis (net benefit) or explicit misclassification costs, we will extract and summarise without imputing costs; multiple author-reported cost scenarios will be presented as sensitivity analyses. [37–39]

### b. Outcomes & Synthesis

Primary metric & pooling: only ROC-AUC will be meta-analysed (random-effects on logit-AUC). Pooling requires ≥5 clinically comparable studies (same target, prediction horizon, and validation tier). We summarise heterogeneity with $\tau^2$ and $I^2$; if $I^2 > 75\%$ or subgroups are sparse/incoherent, we will not pool. [40]

Interpretive weighting under imbalance: while only ROC-AUC is pooled, PR-AUC and MCC will receive greater interpretive weight in narrative/visual synthesis for imbalanced datasets. [35,36]

When pooling is inappropriate (e.g., if criteria are unmet, such as sparse subgroups, incompatible outcomes, or overlapping cohorts), we will use a structured narrative following SWiM guidance, accompanied by standardised tables/figures. [40]

### Risk-of-bias and methodological quality

Even though this is a methodological scoping review, we will apply a tailored quality checklist informed by TRIPOD+AI report items and PROBAST/PROBAST-AI domains (focus on reproducibility, data leakage safeguards, validation, and calibration reporting) to describe reporting quality and potential bias. [34]. We will apply design-level screening for reproducibility, influence diagnostics (Cook's distance [41], studentized residuals [42]), and small-study-effect tests (funnel plot [42], Egger's regression [43], and Vevea–Hedges' weight function [44]) to inform sensitivity analyses. We will continue to assess whether studies report blinding, handle missing data, and provide external validation; we plan to incorporate these elements into a supplementary risk-of-bias table. Although methodological, we will apply a tailored checklist drawing on TRIPOD+AI (reporting) and PROBAST/PROBAST+AI domains (risk of bias/applicability). Results summarised narratively (no scoring).

### Terminology and bias signals

To avoid ambiguity, we standardise terminology and use resampling strategy to denote data-level methods (random over-/undersampling, SMOTE variants, hybrids). We adopt the term "small-study effects" as an umbrella term for patterns whereby smaller studies report larger effects; such patterns can arise from publication bias, outcome-reporting bias, lower study quality, between-study heterogeneity, or chance. We will inspect funnel plot asymmetry and, where feasible, apply Egger's test as a *screening* tool. Still, we will interpret asymmetry as evidence of small-study effects, rather than publication bias alone, and discuss plausible causes in context. [45,46]

### How we'll report it

Consistent with PRISMA 2020, we will report whether small-study effects were assessed, which methods were used (visual inspection, Egger's test), and limitations of these tests. We will refrain from formal testing when subgroups contain too few studies (e.g., <10), and will emphasise qualitative interpretation when power is low.

### Data synthesis

Phase 1—Descriptive mapping: Tables and visualisations (e.g., heat maps, temporal plots) will summarise trends in resampling use, model type, imbalance severity, and performance.

Phase 2 — Quantitative synthesis: Random-effects meta-regression of logit-AUC will examine moderators (imbalance ratio, sample size, resampling strategy, model family). Pooling requires ≥5 clinically coherent studies (same target, horizon, validation tier). The REML estimator and Knapp-Hartung confidence intervals will be employed [42]. Heterogeneity will be assessed using $\tau^2$ and $I^2$ [42]; leave-one-out analyses will be used to test robustness. The analyses will be implemented in R (metafor, dplyr, ggplot2) [42]. If $I^2$ is very high (≈>75%) or subgroups are sparse/incoherent, we will not pool and will follow SWiM for structured narrative synthesis. [40]

## Subgroup and sensitivity analyses

Planned subgroup contrasts include oversampling versus undersampling, hybrid versus single-technique pipelines, cost-sensitive versus data-level only, high (>20%) versus very low (<5%) minority prevalence, and deep learning versus traditional models. Sensitivity analyses will exclude studies with high influence, those lacking external validation, and studies without calibration reporting. The imbalance ratio (IR) will be stratified a priori into four bins: very rare (< 5%), rare (5–10%), moderate (10–20%), and mild (20–30%) [6]. If any bin contains fewer than 10 studies, it will be merged with the next wider bin. For meta-regression, these bins will be dummy-coded (reference = mild), and IR will also be modelled as a restricted cubic spline to test linearity. Sensitivity analyses will replicate the model using two dichotomies (< 10% versus ≥ 10%; < 20% versus ≥ 20%).

## Living review plan

Given the rapid methodological advances, automated database alerts will rerun the search on an annual basis. New eligible studies will be screened and, where appropriate, integrated into updated meta-analyses, with a version history logged transparently.

## Discussion

Class imbalance remains one of the most persistent threats to safe clinical prediction: skewed data encourages algorithms to optimise overall accuracy at the expense of rare—but clinically essential—events. Algorithm-level approaches that embed explicit misclassification penalties can theoretically offset this bias [47,48]. Simultaneously, recent deep learning innovations such as deep belief networks and focal loss functions promise further gains in high-dimensional settings [49,50]. However, the empirical value of these strategies has never been systematically synthesised across the medical spectrum. Our planned scoping review with meta-regression addresses a critical methodological gap.

## Anticipated challenges

- **Extreme heterogeneity:** preliminary scoping indicates a broad dispersion in clinical domains, imbalance ratios, sample sizes, and metrics. Even when studies report AUC, converting to a common logit scale may not entirely harmonise differences in test–set construction and cross-validation folds.

- **Inconsistent reporting:** fewer than one in ten studies in the initial screening publish calibration indices, and details of cost-sensitive losses are frequently relegated to supplementary code or omitted entirely.

- **Sparse external validation**: Most papers evaluate performance using random internal splits; true generalisability remains uncertain.

- **Publication and small-study effects:** Funnel plot asymmetry is anticipated, as smaller datasets often utilise aggressive oversampling, which skews apparent discrimination.

- **Metric multiplicity**: sensitivity, specificity, F-score, precision-recall AUC, and balanced accuracy are reported idiosyncratically, complicating quantitative synthesis.

## Strengths

- **The breadth of evidence:** includes five bibliographic databases and grey literature repositories, encompassing 15 years of work and yielding the most extensive curated corpus of imbalance-related prediction studies.

- **Dual synthesis:** A descriptive map is paired with a random-effects meta-regression that explores moderators such as imbalance severity, sample size, and model family, yielding detailed insights not available in narrative reviews.

- **Rigorous bias diagnostics**: Influence statistics, funnel-plot tests, trim-and-fill, and Vevea–Hedges models will quantify the robustness of pooled estimates, alleviating the optimism that pervades the model-development literature.

- **Technology-enabled workflows**: ML-assisted screening using ASReview accelerates and transparently documents selection decisions [33].

- **Alignment with contemporary guidance**: Search, extraction, and reporting follow the PRISMA 2020 extensions to enhance reproducibility and uptake [25].

## Limitations

Despite these safeguards, several constraints persist. First, residual heterogeneity is unavoidable; even a comprehensive meta-regression may elucidate only a modest fraction of the variance observed between studies. Second, using AUC as the primary effect size may risk neglecting threshold-dependent performance and real-world applications. Third, cost-sensitive studies might still be too few or inconsistently reported to facilitate quantitative pooling, necessitating a descriptive approach that limits formal comparisons through resampling. Fourth, living-review updates will depend on the speed at which newly published work reports compatible statistics—the review may lag very recent methodological advances.

## Potential impact and influence on practice

By determining when and for whom resampling or weighting truly adds value, this review will assist data scientists in avoiding reflexive oversampling, which can obscure calibration or encourage overfitting. Evidence suggests that cost-sensitive losses rival data-level balancing, which could shift practice towards simpler, loss–function–centric pipelines readily available in mainstream frameworks [47–50]. Clinicians and journal editors might utilise the findings to demand more comprehensive reporting of calibration, confusion matrices, and misclassification costs, thereby accelerating the adoption of emerging AI reporting extensions (e.g., TRIPOD-AI), see also [25]. Regulators may likewise refer to our recommendations when evaluating the fairness of deployed diagnostic or prognostic models.

## Future directions

The mapped gaps suggest four priorities:

1. **Prospective, multi-centre cohorts with rare outcomes** to test whether cost-sensitive and focal-loss networks outperform oversampling in truly out-of-sample settings.

2. **Standardised reporting templates** that mandate disclosure of class distribution, sampling strategy, calibration, and decision-curve analysis; our findings can feed directly into upcoming guideline revisions.

3. **Generative augmentation and domain-adapted GANs**: Early evidence (e.g., synthetic EEG and radiology data) hints at privacy-preserving promise but requires rigorous external validation [51].

4. **Continuous evidence surveillance** through annual database alerts and semi-automated screening pipelines aligns with the living-review paradigm and ensures the conclusions remain current as new imbalance-handling techniques emerge [25,33].

The planned review will quantify the performance lift (or degradation) attributable to balancing strategies and outline a research agenda for more reproducible, cost-aware, and clinically grounded predictive modelling.

## Supporting information

**S1 File. INPLASY Protocol.** INPLASY Protocol Registration.
(DOCX)

**S2 File. Search Queries.** Ready-to-paste search queries with limit (2009–31$^{st}$ of Dec 2024).
(DOCX)

**S3 File. PRISMA-P Checklist.** PRISMA-P Checklist.
(DOCX)

## Author contributions

**Conceptualization:** Osama Abdelhay.

**Investigation:** Adam Shatnawi.

**Methodology:** Osama Abdelhay, Adam Shatnawi, Hassan Najadat.

**Resources:** Taghreed Altamimi.

**Software:** Adam Shatnawi.

**Supervision:** Osama Abdelhay, Hassan Najadat.

**Validation:** Taghreed Altamimi.

**Writing – original draft:** Osama Abdelhay, Adam Shatnawi, Taghreed Altamimi.

**Writing – review & editing:** Hassan Najadat, Taghreed Altamimi.

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
