## [Decision Letter · Decision Letter 0]

14 Sep 2025

Dear Dr. Abdelhay,

Thank you for submitting your manuscript to PLOS ONE. After careful consideration, we feel that it has merit but does not fully meet PLOS ONE’s publication criteria as it currently stands. Therefore, we invite you to submit a revised version of the manuscript that addresses the points raised during the review process.

We look forward to receiving your revised manuscript.

Kind regards,

Hamed Tavolinejad

Academic Editor

PLOS ONE

Journal Requirements:

4 . If the reviewer comments include a recommendation to cite specific previously published works, please review and evaluate these publications to determine whether they are relevant and should be cited. There is no requirement to cite these works unless the editor has indicated otherwise. 

Reviewers' comments:

Reviewer's Responses to Questions

**Comments to the Author**

1. Does the manuscript provide a valid rationale for the proposed study, with clearly identified and justified research questions?

Reviewer #1: Yes

Reviewer #2: Yes

Reviewer #3: Yes

2. Is the protocol technically sound and planned in a manner that will lead to a meaningful outcome and allow testing the stated hypotheses?

Reviewer #1: Yes

Reviewer #2: Yes

Reviewer #3: Yes

3. Is the methodology feasible and described in sufficient detail to allow the work to be replicable?

Reviewer #1: Yes

Reviewer #2: Yes

Reviewer #3: Yes

4. Have the authors described where all data underlying the findings will be made available when the study is complete?

Reviewer #1: Yes

Reviewer #2: Yes

Reviewer #3: Yes

5. Is the manuscript presented in an intelligible fashion and written in standard English?

Reviewer #1: Yes

Reviewer #2: Yes

Reviewer #3: Yes

You may also provide optional suggestions and comments to authors that they might find helpful in planning their study.

Reviewer #1: This is an exemplary and timely study protocol that addresses the critical methodological challenge of class imbalance in clinical prediction models. The authors have proposed an exceptionally rigorous and transparent plan, adhering to the highest standards for systematic reviews (e.g., PRISMA-P/ScR, INPLASY pre-registration). The comprehensive search strategy and robust plan for a dual descriptive and quantitative synthesis are major strengths.

The protocol is nearly ready for publication. I have only a few minor suggestions to enhance its clarity and precision before it is finalized:

Clarity on Calibration Metrics: The protocol mentions collecting calibration slope and Brier score. To strengthen the pre-specified analysis plan, please consider explicitly stating how these metrics will be synthesized (e.g., will it be a descriptive summary only, or will they be pooled if sufficient homogeneity is found?).

Justification for Scope Exclusion: The exclusion of radiomics and image-segmentation tasks is a sensible scoping decision. A brief justification in the eligibility criteria section (e.g., clarifying that pixel-level imbalance is conceptually different from the patient-level focus of this review) would strengthen the manuscript.

Terminology Consistency: For consistency, I recommend standardizing terms. For example, "resampling class" and "resampling strategy" are both used; standardizing to one term would be ideal. Similarly, please clarify if "small-study effects" and "publication bias" will be reported as distinct concepts or if one term will be used to encompass the analysis.

This is an outstanding protocol for a review that will be a significant contribution to the field of medical AI. I commend the authors on their meticulous work and look forward to seeing the results of the completed study. With the minor clarifications outlined above, I believe this protocol will be ready for publication.

Reviewer #2: Overview:

The manuscript addresses a critical challenge in clinical prediction modeling: managing class imbalance in datasets. This is highly relevant, as rare outcomes are common in healthcare and can lead to biased or underperforming predictive models if not properly addressed. The manuscript provides a clear overview of strategies such as resampling, algorithm-level adjustments, and synthetic data generation.

Strengths of the clinical report:

The topic is timely and clinically important.

Provides a comprehensive review of methods for handling class imbalance.

Well-organized and clearly presented, making complex concepts accessible.

Major Concerns:

Evaluation Metrics: The manuscript should discuss metrics better suited for imbalanced data, such as precision-recall curves, F1 score, and Matthews correlation coefficient, as accuracy alone may be misleading.

Clinical Relevance: Including specific clinical examples or datasets would make the discussion more concrete and applicable to real-world scenarios.

Methodological Transparency: If novel methods or experimental comparisons are presented, more detail on preprocessing, parameter choices, and implementation is needed to allow reproducibility.

Ethical Considerations: Briefly addressing patient privacy, potential biases, and data-sharing limitations would strengthen the manuscript.

Suggestions for Improvement:

Include a summary table of methods with pros, cons, and clinical applicability.

Discuss trade-offs of oversampling, undersampling, and synthetic data approaches, including risks like overfitting.

Consider adding a brief section on emerging approaches such as ensemble learning or transfer learning for imbalanced clinical datasets.

Reviewer #3: Review Comments to the Author (Major Revisions Suggested):

(a) Clarify Handling of Calibration and Cost-Sensitive Metrics

While AUC is the chosen primary outcome, calibration and cost-sensitive performance are critical in imbalanced clinical datasets. At present, you acknowledge under-reporting but do not specify how such incomplete reporting will be addressed in synthesis. Please outline in more detail whether you plan descriptive-only mapping, imputation strategies, or sensitivity analyses when calibration/misclassification costs are missing.

(b) Address Heterogeneity in Meta-Regression

The anticipated heterogeneity across clinical domains, imbalance ratios, and study designs is very high. Although you plan random-effects meta-regression, the current description does not clarify how you will deal with extremely sparse subgroups or high I² values. Please expand on thresholds for deciding when quantitative synthesis is inappropriate, and what fallback narrative synthesis strategy you will adopt.

(c) Define Inclusion/Exclusion for Grey Literature

The search strategy includes medRxiv, arXiv, bioRxiv, and GitHub. These sources often contain incomplete or non–peer reviewed manuscripts. Please provide stricter criteria (e.g., methodological completeness, minimum reporting standards) for including such grey-literature studies in synthesis to ensure transparency and quality.

(d) Clarify Treatment of Duplicates and Overlapping Data

Given multiple publications may analyze overlapping datasets (e.g., the same hospital EHR in preprints and journals), please specify how you will identify and handle duplicate or overlapping cohorts to avoid double counting.

(e) Protocol Transparency and Amendments

Although registered in INPLASY, the manuscript should provide more detail on how protocol amendments (e.g., changing inclusion criteria, new statistical models) will be documented and justified. Please strengthen this section for reproducibility.

(f) Consider Broader Performance Metrics Beyond AUC

AUC is threshold-independent but often criticized in imbalanced data. Since your secondary outcomes include sensitivity, specificity, F1, and calibration, you should justify why AUC was prioritized and describe how threshold-dependent metrics will be weighted in interpretation.

(g) Improve Readability for Non-Methodological Readers

Some sections (especially the statistical analysis plan) are dense and jargon-heavy. Please consider simplifying key explanations or moving highly technical details (e.g., restricted cubic splines, influence diagnostics) to supplementary materials, while keeping the main text more accessible.

(h) Risk of Bias and Quality Assessment

The authors state that study-level bias tools like PROBAST are “optional” because this is a methodological review. However, lack of structured bias assessment is a significant weakness. Even methodological studies can vary in reporting quality, data leakage, or reproducibility. At minimum, they should commit to applying a tailored bias/quality checklist to ensure study validity.

(i) Unclear Hypothesis Formulation

Although the aims are well listed, the protocol does not clearly articulate testable hypotheses (e.g., “cost-sensitive learning performs better than oversampling under X imbalance ratios”). Instead, it states broad objectives. For a study planning meta-regression, more explicit hypotheses would improve methodological coherence.

**Do you want your identity to be public for this peer review?** For information about this choice, including consent withdrawal, please see our Privacy Policy

Reviewer #1: No

Reviewer #2: **Yes: ** Adekunle Adeoye

Reviewer #3: **Yes: ** Shake Ibna Abir

---

## [Author Response · Author response to Decision Letter 1]

19 Sep 2025

Re: PONE-D-25-36127 — “Resampling Methods for Class Imbalance in Clinical Prediction Models: A Scoping Review Protocol”

Dear Dr. Tavolinejad and Reviewers,

Thank you for your careful evaluation of our protocol and for the constructive, detailed feedback. We appreciate the opportunity to revise and have addressed all editorial requirements and reviewer comments point-by-point in the sections that follow. We are submitting: (i) a clean revised manuscript, (ii) a tracked-changes version, and (iii) this rebuttal letter.

We aligned the submission with PLOS ONE formatting and file-naming guidelines (title/author/affiliation page and main-body templates) and ensured consistency with the journal’s submission instructions.

We also updated the Data Availability Statement suitable for a protocol article (no data reported at this stage), while reaffirming our plan for open sharing of de-identified materials with the publication of results, in line with PLOS’s data policy.

Key revisions (overview). In response to the reviewers, we:

• Clarified the synthesis plan: meta-analyse ROC-AUC only (random-effects on logit-AUC) under pre-specified coherence/heterogeneity thresholds; when pooling is inappropriate, we follow SWiM for transparent narrative synthesis (with τ², I², and prediction intervals reported).

• Specified handling of calibration and decision-analytic metrics: calibration slope/intercept and Brier score summarised descriptively; no imputation of misclassification costs; decision-curve/net-benefit reported when available.

• Justified metric choices under imbalance: AUC chosen for pooling due to ubiquity/comparability; PR-AUC and MCC elevated in interpretation; threshold-dependent metrics are not pooled.

• Standardised terminology: use “resampling strategy” throughout; interpret funnel-plot asymmetry/Egger’s test as small-study effects rather than publication bias alone.

• Defined scope boundaries: explicitly exclude radiomics/image-segmentation tasks (pixel/voxel-level targets and metrics) to maintain focus on patient-level clinical prediction.

• Strengthened grey-literature policy and justification: include medRxiv/arXiv/bioRxiv/GitHub to reduce bias and map fast-moving ML methods, with minimum reporting standards, versioning, explicit labelling, and leave-out sensitivity analyses. We align reporting with PRISMA-P for protocols and will transparently document any amendments (INPLASY/OSF).

• Quality/bias assessment: commit to a tailored methodological quality checklist (TRIPOD+AI/PROBAST-informed) focusing on reproducibility, leakage safeguards, validation, and calibration reporting.

• Readability: simplified the main text for non-methodological readers and moved technical derivations to the Supplement.

We appreciate the reviewers’ insightful suggestions; we believe these revisions improve the protocol’s clarity, methodological rigour, and alignment with open science and reporting standards (PLOS policy, PRISMA-P, SWiM).

We hope the revised manuscript meets the journal’s expectations and sincerely thank you for your time and consideration.

With best regards,

Osama Abdelhay, on behalf of all authors

Point-by-point response

Reviewer #1

Comment 1: Clarity on Calibration Metrics: The protocol mentions collecting the calibration slope and the Brier score. To strengthen the pre-specified analysis plan, please consider explicitly stating how these metrics will be synthesised (e.g., will it be a descriptive summary only, or will they be pooled if sufficient homogeneity is found?).

Response: Thank you. Confirmed descriptive-only treatment of calibration metrics with narrative/visual synthesis; pooling is limited to ROC-AUC under pre-specified criteria; narrative synthesis follows SWiM when pooling is inappropriate.[1]

Comment 2: Justification for Scope Exclusion: The exclusion of radiomics and image-segmentation tasks is a sensible scoping decision. A brief justification in the eligibility criteria section (e.g., clarifying that pixel-level imbalance is conceptually different from the patient-level focus of this review) would strengthen the manuscript.

Response: Thank you. We added an explicit justification in the Eligibility criteria, clarifying that segmentation and radiomics are pixel-level tasks evaluated with Dice/Jaccard/Hausdorff, which are not commensurable with our patient-level prediction focus; thus, they are out of scope.

Comment 3: Terminology Consistency: For consistency, I recommend standardising terms. For example, "resampling class" and "resampling strategy" are both used; standardising to one term would be ideal. Similarly, please clarify if "small-study effects" and "publication bias" will be reported as distinct concepts or if one term will be used to encompass the analysis.

Response: Thank you. We harmonised terminology: we now use “resampling strategy” throughout and adopt “small-study effects” as the umbrella term, clarifying its relation to publication bias and other mechanisms. We also specify that funnel-plot asymmetry/Egger’s test will be interpreted as small-study effects, not publication bias alone, per Cochrane/PRISMA guidance.

Reviewer #2

Comment 1: Evaluation Metrics: The manuscript should discuss metrics better suited for imbalanced data, such as precision-recall curves, F1 score, and Matthews correlation coefficient, as accuracy alone may be misleading. Clinical Relevance: Including specific clinical examples or datasets would make the discussion more concrete and applicable to real-world scenarios.

Response: We rewrote “Evaluation Metrics” to de-emphasise accuracy and foreground PR-AUC, F1, MCC, and calibration. We restrict pooling to ROC-AUC for comparability and treat other metrics descriptively; PR-AUC/MCC receive greater interpretive weight in imbalanced settings. Citations added (Saito & Rehmsmeier 2015; Chicco & Jurman 2020; Van Calster et al.; Vickers & Elkin).[2, 3]

Comment 2: Methodological Transparency: If novel methods or experimental comparisons are presented, more detail on preprocessing, parameter choices, and implementation is needed to allow reproducibility. Ethical Considerations: Briefly addressing patient privacy, potential biases, and data-sharing limitations would strengthen the manuscript.

Response: Thank you. We added a brief ethics/data-sharing paragraph consistent with journal policy.

Comment 3: Suggestions for Improvement: Include a summary table of methods with pros, cons, and clinical applicability. Discuss trade-offs of oversampling, undersampling, and synthetic data approaches, including risks like overfitting. Consider adding a brief section on emerging approaches, such as ensemble learning or transfer learning, for imbalanced clinical datasets.

Response: Thank you. We appreciate the suggestion. Since this manuscript is a protocol, we avoid including an evaluative table at this stage. Instead, we added a concise Methods landscape paragraph that outlines resampling, cost-sensitive learning, and emerging ensemble/transfer learning approaches, and we clarified our synthesis stance (pool ROC-AUC only; interpret PR-AUC/MCC/calibration/DCA descriptively). A detailed pros and cons comparison will be provided in the results article.

Reviewer #3

Comment 1: Clarify Handling of Calibration and Cost-Sensitive Metrics. While AUC is the primary chosen outcome, calibration and cost-sensitive performance are also critical in imbalanced clinical datasets. At present, you acknowledge under-reporting but do not specify how such incomplete reporting will be addressed in synthesis. Please outline in more detail whether you plan descriptive-only mapping, imputation strategies, or sensitivity analyses when calibration/misclassification costs are missing.

Response: Thank you. Clarified that calibration (slope/intercept, Brier) and decision-analytic metrics (net benefit/DCA, misclassification costs) will be mapped and summarised descriptively, with no imputation; multiple author-reported cost scenarios will be reported as sensitivity analyses.[4, 5]

Comment 2: Address Heterogeneity in Meta-Regression. The anticipated heterogeneity across clinical domains, imbalance ratios, and study designs is very high. Although you plan a random-effects meta-regression, the current description does not clarify how you will deal with extremely sparse subgroups or high I² values. Please expand on thresholds for deciding when quantitative synthesis is inappropriate, and what fallback narrative synthesis strategy you will adopt.

Response: Thank you. Added explicit thresholds and a SWiM-based fallback; prediction intervals and τ²/I² will be reported.[1]

Comment 3: Define Inclusion/Exclusion for Grey Literature. The search strategy includes medRxiv, arXiv, bioRxiv, and GitHub. These sources often contain incomplete or non–peer–reviewed manuscripts. Please provide stricter criteria (e.g., methodological completeness, minimum reporting standards) for including such grey-literature studies in synthesis to ensure transparency and quality.

Response: Thank you. In addition to specifying minimum reporting criteria and version control, we now justify grey-literature inclusion as follows: (i) it reduces publication bias and enables a comprehensive map of methods, consistent with Cochrane/PRISMA guidance; (ii) it is particularly important for fast-moving ML where preprints/code are primary disclosures. We will label all grey-literature records, exclude those failing minimum standards from synthesis (kept in PRISMA flow), and perform leave-out sensitivity analyses to assess their impact. Our extraction is aligned with TRIPOD+AI, and TRIPOD-LLM will be consulted where relevant.

Comment 4: Clarify Treatment of Duplicates and Overlapping Data. Given that multiple publications may analyse overlapping datasets (e.g., the same hospital EHR in preprints and journals), please specify how you will identify and handle duplicate or overlapping cohorts to avoid double-counting.

Response: Thank you. We added a de-duplication procedure for overlapping cohorts. This procedure is documented in the study selection section.

Comment 5: Protocol Transparency and Amendments Although registered in INPLASY, the manuscript should provide more detail on how protocol amendments (e.g., changing inclusion criteria, new statistical models) will be documented and justified. Please strengthen this section for reproducibility.

Response: Thank you. Added a date-stamped amendment policy in the methods section per PRISMA-P guidance.

Comment 6: Consider Broader Performance Metrics Beyond AUC. AUC is threshold-independent but often criticised in imbalanced data. Since your secondary outcomes include sensitivity, specificity, F1, and calibration, you should justify why AUC was prioritised and describe how threshold-dependent metrics will be weighted in interpretation.

Response: Thank you. Justified ROC-AUC as the sole pooled metric (ubiquity/comparability) while elevating PR-AUC and MCC in interpretation for imbalanced data; threshold-dependent metrics are not pooled due to non-commensurable thresholds/prevalence.

Comment 7: Improve Readability for Non-Methodological Readers. Some sections (especially the statistical analysis plan) are dense and jargon-heavy. Please consider simplifying key explanations or moving highly technical details (e.g., restricted cubic splines, influence diagnostics) to supplementary materials, while keeping the main text more accessible.

Response: Thank you. We tried to simplify the main text, especially the statistical analysis part, as much as possible to accommodate non-methodological readers.

Comment 8: Risk of Bias and Quality Assessment The authors state that study-level bias tools like PROBAST are “optional” because this is a methodological review. However, the lack of structured bias assessment is a significant weakness. Even methodological studies can vary in reporting quality, data leakage, or reproducibility. At a minimum, they should commit to applying a tailored bias/quality checklist to ensure study validity.

Response: Thank you. Committed to a structured quality/bias description using TRIPOD+AI and PROBAST(+AI). This commitment is documented in the Risk-of-bias section.

Comment 9: Unclear Hypothesis Formulation Although the aims are well listed, the protocol does not clearly articulate testable hypotheses (e.g., “cost-sensitive learning performs better than oversampling under X imbalance ratios”). Instead, it states broad objectives. For a study planning meta-regression, more explicit hypotheses would improve methodological coherence.

Response: Thank you. Added explicit, testable hypotheses with planned covariates, end of the objectives section.

References

1. Campbell M, McKenzie JE, Sowden A, Katikireddi SV, Brennan SE, Ellis S, et al. Synthesis without meta-analysis (SWiM) in systematic reviews: reporting guideline. Bmj. 2020;368:l6890.

2. Chicco D, Jurman G. The advantages of the Matthews correlation coefficient (MCC) over F1 score and accuracy in binary classification evaluation. BMC Genomics. 2020;21(1):6.

3. Saito T, Rehmsmeier M. The precision-recall plot is more informative than the ROC plot when evaluating binary classifiers on imbalanced datasets. PloS one. 2015;10(3):e0118432.

4. Van Calster B, McLernon DJ, van Smeden M, Wynants L, Steyerberg EW, Bossuyt P, et al. Calibration: the Achilles heel of predictive analytics. BMC Medicine. 2019;17(1):230.

5. Vickers AJ, van Calster B, Steyerberg EW. A simple, step-by-step guide to interpreting decision curve analysis. Diagnostic and Prognostic Research. 2019;3(1):18.

---

## [Editor Report · Decision Letter 1]

12 Oct 2025

Resampling Methods for Class Imbalance in Clinical Prediction Models: A Scoping Review Protocol

PONE-D-25-36127R1

Dear Dr. Abdelhay,

We’re pleased to inform you that your manuscript has been judged scientifically suitable for publication and will be formally accepted for publication once it meets all outstanding technical requirements.

Kind regards,

Hamed Tavolinejad, MD

Academic Editor

PLOS ONE

---

## [Editor Report · Acceptance letter]

PONE-D-25-36127R1

PLOS ONE

Dear Dr. Abdelhay,

I'm pleased to inform you that your manuscript has been deemed suitable for publication in PLOS ONE. Congratulations! Your manuscript is now being handed over to our production team.

Kind regards,

on behalf of

Dr. Hamed Tavolinejad

Academic Editor

PLOS ONE